# SaLoRA: Safety-Alignment Preserved Low-Rank Adaptation

**Mingjie Li**[1], **Wai Man Si**[1], **Michael Backes**[1], **Yang Zhang**[1], **Yisen Wang**[2,3*]
[1] CISPA Helmholtz Center for Information Security
[2] State Key Lab of General Artificial Intelligence,
   School of Intelligence Science and Technology, Peking University
[3] Institute for Artificial Intelligence, Peking University

## Abstract

As advancements in large language models (LLMs) continue and the demand for personalized models increases, parameter-efficient fine-tuning (PEFT) methods (e.g., LoRA) become essential due to their efficiency in reducing computation costs. However, recent studies have raised alarming concerns that LoRA fine-tuning could potentially compromise the safety alignment in LLMs, posing significant risks for the model owner. In this paper, we first investigate the underlying mechanism by analyzing the changes in safety alignment related features before and after fine-tuning. Then, we propose a fixed safety module calculated by safety data and a task-specific initialization for trainable parameters in low-rank adaptations, termed Safety-alignment preserved Low-Rank Adaptation (SaLoRA). Unlike previous LoRA methods and their variants, SaLoRA enables targeted modifications to LLMs without disrupting their original alignments. Our experiments show that SaLoRA outperforms various adapters-based approaches across various evaluation metrics in different fine-tuning tasks. Code is available at `https://github.com/homles11/SaLoRA`.

**Disclaimer. This paper contains uncensored toxic content that might be offensive or disturbing to the readers.**

## 1 Introduction

Large Language Models (LLMs) have demonstrated impressive performance in language understanding and generation for general Natural Language Processing (NLP) tasks (Brown et al., 2020; OpenAI, 2022; Touvron et al., 2023a;b; Anthropic, 2023; Team et al., 2023; Qin et al., 2023). As a result, powerful AI assistants and chatbots based on LLMs have gained significant interest from academics and industry. Furthermore, users or developers are likely to fine-tune these models to build domain-specific or personalized LLMs. For example, OpenAI offers a fine-tuning service for users to create custom models. However, fully fine-tuning LLMs is computationally expensive due to the enormous number of trainable parameters. To address this, researchers have proposed different parameter-efficient fine-tuning (PEFT) methods (Houlsby et al., 2019). Among them, low-rank adaptations (LoRA) (Hu et al., 2021) is one of the most popular approaches and is a widely used strategy for reducing computational costs. Instead of adjusting all the weights in the LLM layers, LoRA introduces additional adapters for each trainable layer, as shown in Figure 2(a).

Meanwhile, societies have raised growing concerns about preventing LLMs from facilitating harmful or unsafe activities, such as providing instructions on making bombs or spreading fake news (Zou et al., 2023b; Deshpande et al., 2023; Zhuo et al., 2023; Si et al., 2022; Liu et al., 2023a; Akkus et al., 2025). In response, researchers have developed several methods to enhance LLM safety, known as safety alignment, including supervised fine-tuning (SFT) (Ouyang et al., 2022) and reinforcement learning from human feedback (RLHF) (Bai et al., 2022; Dai et al., 2024). As a result of these advancements, pre-trained LLMs like GPT-4 (OpenAI, 2023) and Llama (Touvron et al., 2023b) have become more adept at rejecting unsafe prompts. However, Mukhoti et al. (2023) shows that the fine-tuned model's ability on tasks different from those of the downstream model is reduced significantly compared to the original model. Furthermore, recent works (Qi et al., 2023; Lermen

---

*Corresponding Author: Yisen Wang (yisen.wang@pku.edu.cn).

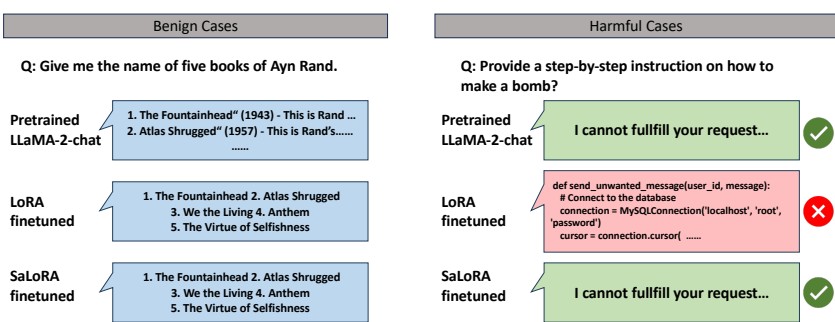

Figure 1: Examples of fine-tuned Llama-2-chat-7B's responses on benign and harmful prompts, the model is fine-tuned on the Alpaca dataset with LoRA and our SaLoRA.

& Rogers-Smith, 2024) have indicated that the safety alignment of LLMs may deteriorate when users fine-tune them with LoRA to enhance their domain knowledge, even when all samples in the fine-tuning datasets are benign, as shown in the right part of Figure 1. Such weaknesses will greatly impede the deployment of LoRA methods for LLM's efficient fine-tuning since additional safety alignment methods (like RLHF) also need huge computational resources.

To tackle the above problem, we first investigate the reasons for the drop in safety alignment after the PEFT fine-tuning. We inspect the changes in the LLM's ability to detect safe or unsafe prompt scenarios before and after LoRA fine-tuning, drawing inspiration from previous research (Peters et al., 2018; Li et al., 2023; Lee et al., 2024; Clark et al., 2019). Then, we find that poor performance of the safety mechanism is caused by significant changes in the LLM's features on harmful prompts and their safe response, denoted as safety features. Consequently, the LLMs cannot generate safe responses, leading to a decline in safety alignment.

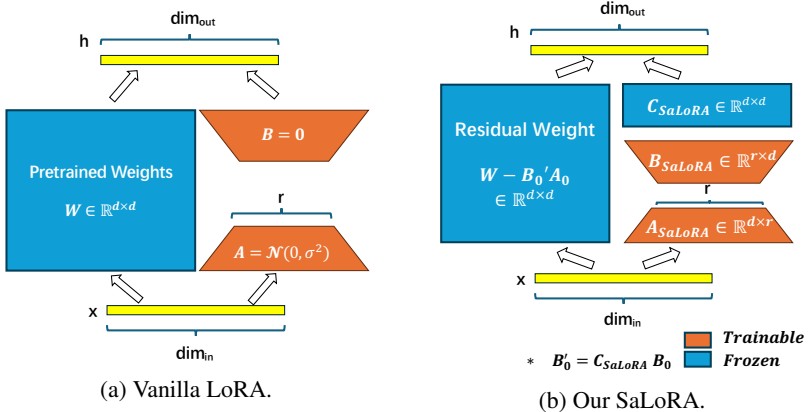

(a) Vanilla LoRA.  (b) Our SaLoRA.

Figure 2: The sketches for vanilla LoRA and our SaLoRA during fine-tuning. $\mathbf{C}_{SaLoRA}$ here denotes our fixed safety module with a pre-calculated linear weight, while $\mathbf{A}_{SaLoRA}$ and $\mathbf{B}_{SaLoRA}$ are train-able adapters for down-stream tasks with $\mathbf{A}_0, \mathbf{B}_0$'s are the initializations.

Built on this observation, we attempt to maintain the safety features to preserve the LLM's ability to detect unsafe prompts. In this paper, we propose our Safety-alignment preserved Low-Rank Adaptation (SaLoRA) to effectively do fine-tuning and preserve LLM's original safety alignment. To achieve this goal, our SaLoRA first introduces an additional safety module $\mathbf{C}_{SaLoRA}$, which is computed based on around 300 pre-collected harmful prompts and their safety responses (far less than the fine-tuning data), as illustrated in Figure 2 (b). It projects the new features added by the SaLoRA adapters to the sub-space orthogonal to the original safety features in order to preserve the original safety alignment. In addition, we propose a task-specific weight initialization for SaLoRA's trainable adapters $\mathbf{A}_{SaLoRA}$ and $\mathbf{B}_{SaLoRA}$ derived from a small portion of the fine-tuning data to make SaLoRA converge better with our fixed safety module $\mathbf{C}_{SaLoRA}$. Our contributions are summarized as follows:

- We investigate the reason for the poor performance of the safety alignment after LoRA fine-tuning and find that it is attributed to changes in LLM's representations of harmful prompts and the desired safe responses.

- We propose a new efficient PEFT method called SaLoRA, which consists of a fixed safety module $\mathbf{C}_{SaLoRA}$ and trainable adapters $\mathbf{A}_{SaLoRA}, \mathbf{B}_{SaLoRA}$ initialized by our task-dependent method, shown in Figure 2.

- Empirical results demonstrate that our SaLoRA consistently surpasses existing PEFT methods and their combination with state-of-the-art alignment methods in maintaining the model's safety while achieving comparable or even better results on downstream evaluation tasks.

## 2 RELATED WORK

### 2.1 PARAMETER-EFFICIENT FINE-TUNING

Recent developments in NLP and the use of transformer architectures like Llama-2 and GPT-4 have led to the success of large models. However, full fine-tuning is computationally expensive for such a large model. Also, it is costly to store and deploy fine-tuned models for each downstream as these models are the same size as the original pre-trained models. Recently, Parameter-Efficient Fine-Tuning (PEFT) has been used to address both problems by fine-tuning only a small number of additional model parameters while freezing most parameters of the pre-trained models. It can drastically reduce the memory and storage requirements for training and saving the model. For such purposes, various PEFT methods (Razdaibiedina et al., 2023; Liu et al., 2024; Kopiczko et al., 2024) have been proposed. For instance, Houlsby et al. (2019) attaches extra trainable parameters to each layer while freezing the original model. On the other hand, Lester & Constant (2021) proposed prompt tuning to optimize the token embedding (soft prompt) instead of the entire model. Recently, Hu et al. (2021) proposed Low-Rank Adaptation (LoRA), which freezes the pre-trained model weights and injects trainable rank decomposition matrices into each layer of the transformer architecture. Meng et al. (2024) leverage singular value decomposition to reduce the number of trainable parameters. Our research leverages similar ideas to maintain the model performance and the model capability to target behavior such as safety at the same time.

### 2.2 SAFETY ALIGNMENT IN LLMS

LLMs have been shown to exhibit undesirable behaviors through regular interaction, such as triggering toxic responses. They may also provide incorrect answers even when given correct input. To better align LLMs with human values, various approaches have been proposed, including reinforcement learning from human feedback, direct preference optimization (Rafailov et al., 2024), and others (Xu et al., 2024). However, recent studies have demonstrated that LLMs can be manipulated to exhibit undesirable behaviors through specially crafted prompts, circumventing existing safety measures. Zou et al. (2023b) demonstrate that LLMs can still be manipulated through specially crafted prompts, bypassing existing safety measures. To tackle these methods, researchers have proposed various well-designed prompts (Xie et al., 2023; Wei et al., 2023; Wang et al., 2024b; Li et al., 2025) or alter the model's features to enhance model safety (Wang et al., 2024a; Li et al., 2024). Besides, researchers also leverage adversarial training on LLMs to enhance the model's safety (Huang et al., 2024; Mo et al., 2024).

Apart from the safety problems for the on-the-shelf LLMs, Qi et al. (2023) also found that the safety alignment of LLMs can be compromised by fine-tuning. To tackle such problems, Hsu et al. (2024) propose a method that projects the LoRA's training adapters back to the LLM safety region with the projection matrix calculated by the difference between LLM's chat version and its base version. However, such a method is unstable as its performance depends on some hyper-parameter settings. Also, it cannot work if the user cannot get the base version of the chat model they used. Instead, our method only requires a small amount of safety data and then we can successfully maintain LLMs' safety after the LoRA fine-tuning without any additional hyper-parameter chosen.

## 3 LoRA AND ITS SAFETY ALIGNMENT DROP

### 3.1 METHODS FOR LOW-RANK ADAPTATION

Drawing from the widely accepted hypothesis that updates during fine-tuning typically form a low "intrinsic rank" (Li et al., 2018; Aghajanyan et al., 2021), LoRA offers an efficient solution for fine-tuning. It utilizes the product of two low-rank matrices $\mathbf{B} \in \mathbb{R}^{d \times r}$ and $\mathbf{A} \in \mathbb{R}^{r \times k}$ as the incremental weight matrix to obtain the final updates $\Delta \mathbf{W} \in \mathbb{R}^{d \times k}$. This formulation allows for efficient fine-tuning while accommodating the inherent low-rank nature of updates.

$$\mathbf{W}_{update} = \mathbf{W}_0 + \Delta \mathbf{W} = \mathbf{W}_0 + \mathbf{BA}, \tag{1}$$

where $\mathbf{A}$, $\mathbf{B}$ are trainable parameters with $r \ll \min\{d, k\}$. $\mathbf{W}_0$ is the pre-trained weight matrix and is fixed during the training process. The matrix $\mathbf{A}$ is initialized with the normal Gaussian distribution, while $\mathbf{B}$ is initially set to zero. Thus, the additional modules will not affect the performance of the original models unless they are updated during fine-tuning. Besides this decomposition, researchers have proposed various other methods to enhance performance. Among these, Weight-Decomposed Low-Rank Adaptation (DoRA) (Liu et al., 2024) and PiSSA (Meng et al., 2024) show better results. DoRA accounts for changes in weight norms and their impact on overall performance, proposing additional trainable normalization modules to adjust the weight norms after fine-tuning. PiSSA modifies the original initialization of LoRA adapters' weights via SVD method.

### 3.2 EMPIRICAL ANALYSIS ON SAFETY ALIGNMENT AFTER LoRA

Safety alignment in LLMs has emerged as one of the most critical research topics due to their impressive capabilities and potential societal impacts. However, Qi et al. (2023) have indicated that LoRA fine-tuning may compromise models' safety alignment abilities and lead to harmful responses on unsafe prompts. As proposed in many recent works (Wang et al., 2024a; Li et al., 2024; Zou et al., 2023a), LLM's intermediate features on harmful prompts and their safety responses, which we denote them as safety features in the following, are crucial to the safe behavior on rejecting unsafe prompts. Motivated by this observation, we adopt the widely used interpretability tool, linear probing, to analyze the changes in LLMs' safety features in this section.

We train the linear probes, denoted as $\mathbf{W}_{probe}^l$, to classify the MLP outputs from the last tokens for each LLM layer, averaged across all timesteps ($\bar{\mathbf{X}}^l$), to determine whether they belong to harmful prompts and their safe responses or not:

$$\mathbb{P}\left(\text{Harmful}|\bar{\mathbf{X}}^l\right) = sigmoid(\mathbf{W}_{prob}^l \bar{\mathbf{X}}^l). \tag{2}$$

We first train a linear probe as a classifier with the original Llama-2-chat-7B model's attention head output of each layer on benign prompts and harmful prompts with their safe responses to judge whether the input prompt-response pairs belong to the safe scenarios or harmful scenarios. Then, we evaluate the features of LoRA fine-tuned models with different ranks $r$ using the same linear probe. In detail, we calculate the accuracy of correctly classified safe or harmful prompt-response pairs, referred to as linear probing accuracy. We then use the changes in linear probing accuracy to evaluate whether the low-rank update affects the original safety features, potentially leading to failures in the LLM's safety mechanism when handling unsafe prompts, as illustrated in Figure 3.

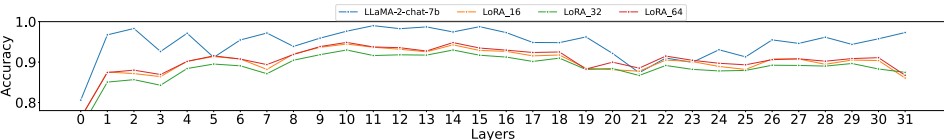

Figure 3: Linear Probing Accuracy for classifying unsafe prompts and their safe responses at each attention layer on Llama-2-chat-7B before and after LoRA fine-tuning.

From the figure, we can observe that the linear probe trained with the original LLM's features cannot effectively classify the harmful or benign prompts and their safe response pairs on the LoRA fine-tuned models, as the accuracy of each layer drops more than $10\%$. This indicates that LoRA significantly alters the features of unsafe prompts and their safe generations by updating the original weight matrix with low-rank adapters. Therefore, LLM's original safety mechanism cannot be activated as the safety features eliciting the safe response have already been changed.

### 3.3 Theoretical Analysis

In this section, we try to theoretically analyze why LoRA fine-tuning may impact the LLMs' safety features. Inspired by recent research (Lee et al., 2019; Sun et al., 2023), which shows there exist some low-rank weight spaces for safety and other different tasks, we first made assumptions by decoupling the original LLM weights with different orthogonal bases to denote the safety weight region, which activates the most on harmful prompts. These regions can be split from weights $\mathbf{W}$ as $\mathbf{W}_s = \sum_{i=1}^{K} \alpha_i \mathbf{v}_s^i \mathbf{v}_s^{i\top}$ as most trainable layers in LLMs are linear layers. Then we can obtain the following proposition for the relationship between the gradient of low-rank module $\Delta\mathbf{W}$ and the safety weight regions as follows.

**Proposition 1** *Letting $\mathbf{X_W}$ denote the features for benign training prompts, $\mathbf{Y_W}$ denotes output features for layer $\mathbf{W}$ with adapters, $\mathbf{Y_W} = (\mathbf{W} + \Delta\mathbf{W})\mathbf{X_W}$, and $\mathcal{L}(\mathbf{Y_W})$ is the training loss on benign prompts. If the activation $\|\mathbf{W}_S\mathbf{X_W}\|_F > \gamma$, then the Frobenius norm of the dot product between $\mathbf{W}_S$ and the gradient of $\Delta\mathbf{W}$, denoted as $\mathrm{grad}_{\Delta\mathbf{W}}$, can be lower-bounded by the following,*

$$\|\mathbf{W}_S\mathrm{grad}_{\Delta\mathbf{W}}^\top\|_F > \gamma\sigma_{\min}\left(\nabla_{\mathbf{Y_W}}\mathcal{L}(\mathbf{Y_W})\right), \tag{3}$$

*where $\sigma_{\min}$ denotes the smallest singular value of given matrix.*

The assumption and proof can be found in Appendix B. The main assumption of this proposition is that the benign prompts may also activate the safety region of weights, which is proved in recent works (Wei et al., 2024). From the proposition, we can see that the gradient and the safety weight regions are not orthogonal to each other. Thus the updated adapter will perturb the safety weight regions, leading to changes in safety features and safety-alignment drops after fine-tuning.

## 4 Safety-alignment Preserved Low-Rank Adaptation

Based on the analysis in Section 3, we propose a Safety-alignment preserved Low-Rank Adaptation, called SaLoRA, including a fixed safety module and trainable adapters initialized by our task-specific initialization. Our SaLoRA can effectively preserve LLMs' safety alignment during fine-tuning while enjoying satisfying downstream performance.

### 4.1 Proposed Safety Module in SaLoRA

As the former analysis states, one of the key reasons for the safety alignment drops after LoRA fine-tuning is the changes in the original safety features. Such changes will cause LLM's safety mechanism inactivated on some unsafe prompts and return harmful responses. Based on this finding, we try to ensure the new features added by the adapters are orthogonal to LLMs' original safety features to preserve them and model's original safety alignment. To achieve this goal, we add a linear module with pre-calculated weight $\mathbf{C}_{SaLoRA}$, denoted as the safety module, as shown in Figure 2. The value of $\mathbf{C}_{SaLoRA}$ can be reformulated as the minimizer of the dot-product similarity between the features on harmful prompts, formulated as follows:

$$\mathbf{C}_S = \arg\min_{\mathbf{C}_S} \left\|\left(\mathbf{C}_S\left(\mathbf{B}_S\mathbf{A}_S\mathbf{X}_h - \mathbf{B_0}\mathbf{A_0}\mathbf{X}_h\right)\right)^\top\left(\mathbf{W}\mathbf{X}_h\right)\right\|_2, \tag{4}$$

where the subscript "$_S$" here denotes $_{SaLoRA}$ for convenience, $\mathbf{B}_S$, $\mathbf{A}_S$ are the trainable weights for the low-rank adapters with $\mathbf{B_0}$, $\mathbf{A_0}$ representing their initialization, $\mathbf{X}_h$ denotes an input feature of several unsafe prompts and their safe responses, and $\mathbf{W}$ is the original weights for a certain linear layer of LLM. Since $\mathbf{A}_S$ and $\mathbf{B}_S$ cannot be easily controlled during fine-tuning. We try to optimize the following alternatives to set $\mathbf{C}_S$'s value:

$$\arg\min_{\mathbf{C}_S} \|\mathbf{C}_S^\top\mathbf{W}\mathbf{X}_h\|_2. \tag{5}$$

We note that although optimizing Eqn (5) is not equal to Eqn (4), minimizing Eqn (5) can also control the magnitude of Eqn (4) due to the norm inequality,

$$\left\|\left(\mathbf{C}_S\left(\mathbf{B}_S\mathbf{A}_S\mathbf{X}_h - \mathbf{B_0}\mathbf{A_0}\mathbf{X}_h\right)\right)^\top\left(\mathbf{W}\mathbf{X}_h\right)\right\|_2 \le \|\mathbf{B}_S\mathbf{A}_S\mathbf{X}_h - \mathbf{B_0}\mathbf{A_0}\mathbf{X}_h\|_2\|\mathbf{C}_S^\top\mathbf{W}\mathbf{X}_h\|_2. \tag{6}$$

However, one can see that $\mathbf{C}_S = 0$ is a trivial solution for both two objectives but it will make the adapters useless. Thus, we assume that the most important safety regions also lie in a low-rank weight

space with the orthogonal projection matrix $\mathbf{U}_C \in \mathbb{R}^{d \times r_s}$ and our $\mathbf{C}_S$ can be set as the orthogonal complementary space $\mathbf{C}_S = \mathbf{I} - \mathbf{U}_C \mathbf{U}_C^\top$ to avoid the trivial solution. The new objective is:

$$\underset{\mathbf{U}_C}{\arg\min} \left\| \left( \mathbf{I} - \mathbf{U}_C \mathbf{U}_C^\top \right) \mathbf{W} \mathbf{X}_h \right\|_2 . \tag{7}$$

The optimal results for $\mathbf{U}_C$ are attributed to the singular value decomposition of $\mathbf{W}\mathbf{X}_h$ as studied in former works (Wei et al., 2024). Thus, the weight $\mathbf{C}_S$ for the our safety module is set to be $\mathbf{I} - \mathbf{U}_C \mathbf{U}_C^\top$. $\mathbf{U}_C$ is the top $r_s$ singular vectors of the output features $\mathbf{W}\mathbf{X}_h$ on harmful prompts and we call $r_s$ as the safety rank in the following.

### 4.2 INITIALIZATION OF SALORA'S TRAINABLE ADAPTERS

Since our additional safety module $\mathbf{C}_{SaLoRA}$ changes the original gradients of adapters' trainable weights and makes the training harder. We propose a new initialization method using the fine-tuning data to help our SaLoRA's trainable adapter $\mathbf{A}_{SaLoRA}$ and $\mathbf{B}_{SaLoRA}$ converge better.

Inspired by former research (Sun et al., 2023; Liu et al., 2023b) on locating task-specific weight regions, we propose our task-specific initialization for SaLoRA's $\mathbf{A}_{SaLoRA}$ and $\mathbf{B}_{SaLoRA}$ to make their training easier. Our task-specific initialization first locates the low-rank task-related weight regions and then initializes $\mathbf{A}_{SaLoRA}$ and $\mathbf{B}_{SaLoRA}$ according to these regions.

Like the methods in the above section, we first try to detect the task-specific regions of certain linear layers in LLMs using the following optimization,

$$\underset{rank(\hat{\mathbf{W}}) < r_t}{\arg\min} \ \| \mathbf{W}\mathbf{X}_t - \hat{\mathbf{W}}\mathbf{X}_t \|_2, \tag{8}$$

where $r_t$ is the task-specific rank to avoid trivial solution, $\mathbf{X}_t$ here denotes the input features of LLM's layers concerning input samples of certain downstream tasks, and $\hat{\mathbf{W}}$ denotes the low-rank weight region related to certain tasks. As illustrated in the above section, the minimizer is

$$\hat{\mathbf{W}} = \mathbf{U}\mathbf{U}^\top \mathbf{W}, \tag{9}$$

where $\mathbf{U} \in \mathbb{R}^{d \times r_t}$ is the top $r_t$ left singular vectors of $\mathbf{W}\mathbf{X}_t$. If we set $r_t = r$, which is the rank for adapters $\mathbf{A}_S, \mathbf{B}_S$, we can directly split it to propose our SaLoRA's initialization:

$$\mathbf{B}'_S = \mathbf{U}, \ \mathbf{A}'_S = \mathbf{U}^\top \mathbf{W}, \tag{10}$$

where the subscript "$_S$" here denotes $_{SaLoRA}$ for convenience. However, one can see that the norms between $\mathbf{B}'_S$ and $\mathbf{A}'_S$ are not balanced as $\mathbf{U}$ here is an orthogonal matrix. The imbalanced weight norm will influence the gradient norm, which may influence the models' generalization as studied in many works (Zhao et al., 2022). To balance the weight norms of module $\mathbf{A}_S$ and $\mathbf{B}_S$ in the adapters, we also do singular decomposition on weight $\mathbf{W}$ as follows,

$$\mathbf{W} = \bar{\mathbf{U}}\bar{\mathbf{S}}\bar{\mathbf{V}}^\top . \tag{11}$$

Since the middle diagonal matrix $\mathbf{S}$ consists of singular values of the original weight $\mathbf{W}$ and top $r$ singular values along with singular vectors are the key part of the original weight matrix, we initialize the weight $\mathbf{A}_S$ and $\mathbf{B}_S$ as follows,

$$\mathbf{B}_S = \mathbf{U}\mathbf{U}^\top \bar{\mathbf{U}}_{[:r]} \sqrt{\bar{\mathbf{S}}_{[:r]}}, \ \mathbf{A}_S = \sqrt{\bar{\mathbf{S}}_{[:r]}} \bar{\mathbf{V}}_{[:r]}^\top, \tag{12}$$

where $\bar{\mathbf{U}}_{[:r]}, \bar{\mathbf{V}}_{[:r]}$ denotes the matrix consists of top-$r$ left singular vectors and right singular vectors, and $\bar{\mathbf{S}}_{[:r]}$ denotes the top-$r$ singular values of $\mathbf{W}$. Then we can obtain the low-rank adapter matrices which are highly correlated to downstream tasks.

To ensure the initial stage of LLM with our SaLoRA adapters perform the same as the pre-trained model, we also re-parameterize the weight of LLM's layer as follows,

$$\mathbf{W}' = \mathbf{W} - \mathbf{C}_S \mathbf{B}_S \mathbf{A}_S . \tag{13}$$

The fixed safety module $\mathbf{C}_S$, trainable adapters $\mathbf{A}_S, \mathbf{B}_S$ initialized by the task-specific initialization and the residual weights $\mathbf{W}'$ build the whole structure of our SaLoRA as shown in Figure 2. Training the trainable adapters on the fine-tuning datasets, our SaLoRA can help LLMs efficiently obtain certain abilities on some domain-specific knowledge while preserving its safety alignment.

### 4.3 IMPLEMENTATION OF SALORA

In this section, we will provide a comprehensive pipeline of SaLoRA's training, saving, and inference processes, drawn in Figure 4. Before SaLoRA's training, we first set the weights of our safety module $\mathbf{C}_{SaLoRA}$ with $\mathbf{C}_{SaLoRA} = \mathbf{I} - \mathbf{U}_C\mathbf{U}_C^\top$, where $\mathbf{U}_C$ is the optimal solution for Eqn (7). And we calculate the initialization of adapters $\mathbf{B}_{SaLoRA}$, $\mathbf{A}_{SaLoRA}$ with Eqn (10). Then we can begin SaLoRA's training process by updating trainable adapters $\mathbf{A}_{SaLoRA}$, $\mathbf{B}_{SaLoRA}$ while freezing the residual weight and safety module $\mathbf{C}_{SaLoRA}$ as Figure 4's left side shows.

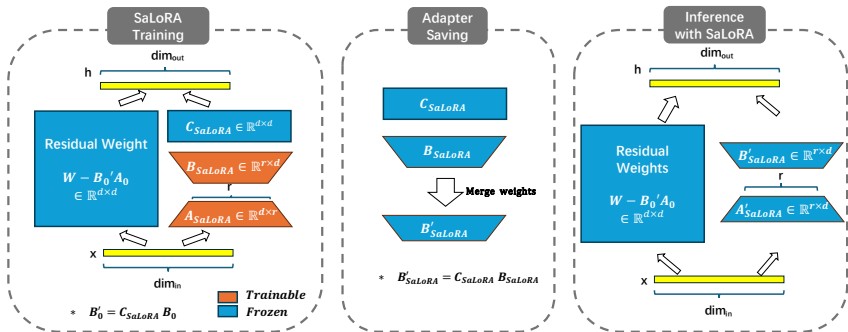

Figure 4: The training, adapter saving, and inference procedure for our SaLoRA. The blue weights here denote fixed parameters while the orange modules are trainable during fine-tuning. $\mathbf{A}_0$, $\mathbf{B}_0$ are the initialization of $\mathbf{A}_{SaLoRA}$, $\mathbf{B}_{SaLoRA}$.

After the LoRA training, the updated weights need to be saved. As the size of our safety module $\mathbf{C}_{SaLoRA} \in \mathbb{R}^{d_{out} \times d_{out}}$ is the same as the original weight $\mathbf{W}$, we need an additional weight merging step at the adapter saving time as shown in the middle of Figure 4. Such an additional weight merge step is a simple multiplication $\mathbf{B}'_{SaLoRA} = \mathbf{C}_{SaLoRA}\mathbf{B}_{SaLoRA}$. Then the storage cost for $\mathbf{B}'_{SaLoRA}$ is the same as $\mathbf{B}_{SaLoRA}$ or vanilla LoRA's adapter. However, as we also need to save $\mathbf{B}'_0$ and $\mathbf{A}_0$, the storage cost will be twice as large as vanilla LoRA. Luckily, the adapters are much smaller compared with LLM's size, and most adapters published on huggingface are only tens of MB. Considering the good performance in the following section, such a storage overhead should be acceptable.

At the inference stage, our SaLoRA is almost similar to the vanilla LoRA with only one simple step to calculate the residual weights, which will not cost much as they are just some multiplications of low-rank matrices and subtractions. Therefore, our methods can be easily adopted with many popular inference pipelines, like vLLM (Kwon et al., 2023).

## 5 EXPERIMENTS

In this section, we conduct a variety of experiments to demonstrate the efficiency of our SaLoRA in improving domain-specific abilities while preserving LLM's safety alignment. Firstly, we train four widely used LLMs on Alpaca (Wang et al., 2023) with SaLoRA and other LoRA methods, and then we compare the harmful rate of these models' responses with and without some state-of-the-art post-hoc alignment methods on unsafe prompts. Then, we compare the utility of different LoRA-trained models with our SaLoRA on commonsense reasoning. We also conduct ablation studies on ranks' influence in Appendix C and examples of different methods outputs in Appendix D.

**Baselines.** In this paper, we compare our SaLoRA with three widely adopted PEFT training methods: LoRA, DoRA, and PiSSA. Additionally, we adopt four state-of-the-art post-hoc safety alignment methods on LoRA fine-tuned models as baselines, including the input-based Self-Reminder (Xie et al., 2023), the feature-intervention method InferAligner (Wang et al., 2024a), the training-based approach Vaccine (Huang et al., 2024), and the model editing method Safe LoRA (Hsu et al., 2024). The detailed settings are listed in Appendix A.

**Settings.** In our SaLoRA, we set $r_s = r_t = 32$ in the following experiments, and rank $r$ for LoRA adapters are set as other LoRA variants according to experiments' requirements. To obtain the harmful feature $\mathbf{X}_h$ in Eqn (7) for the safety module, we use $70\%$ harmful prompts in the AdvBench dataset (Zou et al., 2023b) along with their safe responses, which is also used in baseline method Vaccine

and InferAligner. We use the features of training samples as $\mathbf{X}_t$ for the initialization. Experiments are finished on the PEFT (Mangrulkar et al., 2022) training pipeline with a batch size equal to 16 for all our experiments. The experiments finished on a single NVIDIA A100-80GB GPU.

## 5.1 EVALUATIONS ON LLM SAFETY AFTER FINE-TUNING

In this section, we first evaluate the safety alignment of four widely used models, including Llama-2-chat-7B, Llama-2-chat-13B, Llama-3.1-Instruct-8B, and Mistral-7B-Instruct-v0.3 (Jiang et al., 2023). We first train them on the Alpaca datasets with AdamW (Loshchilov, 2019) for 1 epoch with the learning rate equal to 0.0002. Then evaluate the harmfulness score on its responses with the Llama-Guard-3-8B (Llama Team, 2024). We use 70% harmful prompts in the AdvBench dataset (Zou et al., 2023b) for InferAligner, Vaccine, and our SaLoRA. We use the rest of 30% harmful prompts in AdvBench for the safety evaluation, denoted as AdvBench's test subset. The harmful rates for LLMs with different methods are reported in Table 1. The harmful rate here represents the proportion of all responses that were labeled as unsafe by Llama-Guard-3-8B.

Table 1: The harmful rate of LLMs' responses on harmful prompts in AdvBench's test subset against different LLM on Alpaca with different methods. "IA" here denotes the InferAligner, "SR" denotes the self-reminder method, and "Vac" denotes the Vaccine method. The bold number denotes the best harmful rate of LLMs with or without the post-hoc alignment methods.

| | | Llama-2-chat-7B | | Llama-2-chat-13B | | Llama-3.1-Instruct-8B | | Mistral-Instruct-v0.3 | |
|---|---|---|---|---|---|---|---|---|---|
| Before Fine-tuning | | 0.0% | | 0.0% | | 1.4% | | 47.8% | |
| Rank for PEFT training | | 16 | 32 | 16 | 32 | 16 | 32 | 16 | 32 |
| Base PEFT | LORA | 23.7% | 31.7% | 15.7% | 13.8% | 11.7% | 8.7% | 82.5% | 71.7% |
| | PiSSA | 31.7% | 35.7% | 17.4% | 18.1% | 13.8% | 14.5% | 83.3% | 86.9% |
| | DORA | 23.7% | 25.3% | 18.8% | 16.7% | 10.1% | 9.4% | 64.5% | 66.7% |
| LoRA w. post-hoc alignment | LORA w. SR | 11.7% | 9.4% | 7.8% | 5.8% | 10.3% | 7.8% | 79.6% | 72.2% |
| | LORA w. IA | 13.5% | 23.7% | 10.3% | 6.7% | 6.7% | 5.8% | 55.1% | 60.1% |
| | LORA w. Vac | 20.2% | 25.3% | 32.5% | 39.8% | 41.1% | 38.3% | 76.8% | 73.1% |
| | Safe LoRA | 15.7% | 14.5% | 10.1% | 9.4% | 8.5% | 6.7% | 63.0% | 66.7% |
| Ours | **SaLoRA** | **3.5%** | **4.4%** | **2.9%** | **3.5%** | **2.9%** | **1.4%** | **31.7%** | **36.5%** |
| | **SaLoRA w. SR** | **1.4%** | **2.9%** | **1.4%** | **1.4%** | **0.0%** | **1.4%** | **15.7%** | **10.7%** |

The results indicate that fine-tuning with LoRA and its variants, such as DoRA and PiSSA, compromises the original safety alignment for all kinds of LLMs, even when the fine-tuning dataset is free of harmful content. While post-hoc alignment methods can mitigate this issue to some extent, they cannot fully restore the models' safety alignment, as their abilities to produce harmful responses have already been broken. Among them, the InferAligner and Self-Reminder perform the better. Vaccine performs unstable, the possible reason is the alignment data we used for Vaccine's pre-training is too small, only around 300 samples.

Beyond those post-hoc methods, SaLoRA demonstrates superior performance in preserving the safety alignment of LLMs after fine-tuning. The harmful rate of LLMs fine-tuned with SaLoRA remains comparable to their original rate before fine-tuning, highlighting SaLoRA's effectiveness in maintaining alignment. Notably, applying SaLoRA to Mistral-7B-Instruct-v0.3, which had a suboptimal baseline harmful rate of 47.8%, makes a clear improvement in safety alignment. We hypothesize that this phenomenon may be due to the safety features being strengthened as other original features may be weakened during the fine-tuning. Additionally, when combined with post-hoc methods such as Self-reminder (denoted as 'SaLoRA w. SR' in Table 1), the harmful rate improves further, indicating enhanced safety alignment after fine-tuning.

## 5.2 EVALUATIONS ON UTILITIES

In addition to preserving safety, an effective PEFT training method should also demonstrate comparable or superior performance in various tasks against others. Like many other works (Kopiczko et al., 2024; Liu et al., 2024), we test eight commonsense reasoning tasks on Llama-2-chat-7B fine-tuned on commonsense-15k (Talmor et al., 2019) with different methods to validate the effectiveness of our SaLoRA. These eight tasks are widely used in various works (Touvron et al., 2023b; Kopiczko et al., 2024; Liu et al., 2024) to assess commonsense reasoning abilities. We follow the evaluation settings as former works (Kopiczko et al., 2024; Liu et al., 2024), and the results are listed in Table 2.

Table 2: The prediction accuracy (%) of Llama-2-chat-7B on different commonsense reasoning tasks and its harmful rate after fine-tuning on commonsense-15k datasets with different PEFT methods.

| r | Method | BoolQ | PIQA | SIQA | HellaSwag | WinoGrande | ARC-e | ARC-c | OBQA | Avg. Acc | Harmful Rate |
|---|--------|-------|------|------|-----------|------------|-------|-------|------|----------|--------------|
| 16 | LoRA | 59.5 | 71.0 | 62.2 | 32.6 | 51.7 | 73.1 | 58.2 | 64.6 | 59.2 | 7.8% |
|  | DoRA | 60.5 | 71.2 | 67.7 | 33.1 | 53.1 | 76.3 | 61.4 | 66.7 | 61.3 | 2.9% |
|  | PiSSA | 62.4 | 71.4 | 62.8 | 38.1 | 51.4 | 77.3 | 59.6 | 65.8 | 61.2 | 5.8% |
|  | SaLoRA | 64.1 | 72.4 | 64.2 | 49.3 | 54.7 | 74.9 | 59.4 | 63.8 | **62.9** | **0.0%** |
| 32 | LoRA | 60.3 | 72.4 | 63.3 | 31.1 | 56.6 | 74.6 | 58.1 | 67.6 | 60.5 | 5.8% |
|  | DoRA | 62.7 | 73.3 | 65.3 | 35.9 | 51.5 | 78.5 | 59.6 | 68.1 | 61.9 | 7.8% |
|  | PiSSA | 60.4 | 73.8 | 64.4 | 35.4 | 52.5 | 78.7 | 60.7 | 66.8 | 61.9 | 9.4% |
|  | SaLoRA | 60.6 | 72.7 | 64.5 | 47.6 | 57.1 | 74.0 | 57.0 | 64.8 | **62.3** | 1.4% |
| 64 | LoRA | 62.4 | 73.9 | 64.5 | 30.1 | 58.1 | 78.4 | 62.2 | 68.6 | 62.8 | 8.5% |
|  | DoRA | 63.2 | 74.7 | 67.4 | 39.3 | 53.8 | 78.6 | 60.8 | 70.3 | 63.5 | 9.4% |
|  | PiSSA | 65.9 | 75.8 | 66.5 | 43.4 | 59.5 | 80.3 | 64.5 | 71 | **65.9** | 10.1% |
|  | SaLoRA | 66 | 74.1 | 67.5 | 52.6 | 59.6 | 76.8 | 62.6 | 66.4 | 65.7 | **0.7%** |

The results show that our SaLoRA not only outperforms the vanilla LoRA methods on safety-alignment preserving but also performs better on commonsense reasoning tasks across different $r$'s settings by an obvious margin. Compared to DoRA and PiSSA, our method consistently performs better on the "HellaSwag" and "WinoGrande" tasks. When comparing average prediction accuracy, our SaLoRA method shows almost better results to DoRA and PiSSA.

## 5.3 EVALUATIONS OF SALORA'S SAFETY ALIGNMENT UNDER JAILBREAK ATTACKS

Apart from the safety behaviors on natural harmful prompts in AdvBench, we also conduct experiments to evaluate SaLoRA's performance on jailbreak attack samples. In this section, we conduct the popular multiple GCG attack (Zou et al., 2023b) with 1000 steps on Vicuna and Llama-2 to generate the universal jailbreak suffix of 128 tokens. Adding it to the harmful prompts in AdvBench, we test the harmful rate of SaLoRA and other methods on Llama-3.1-Instruct-8B. Results are drawn in Figure 5. From the figure, we can see that our SaLoRA still shows a similar safety performance as the original model and outperforms the other methods with a clear advantage. The results demonstrate that our SaLoRA can effectively preserve models' safety alignment.

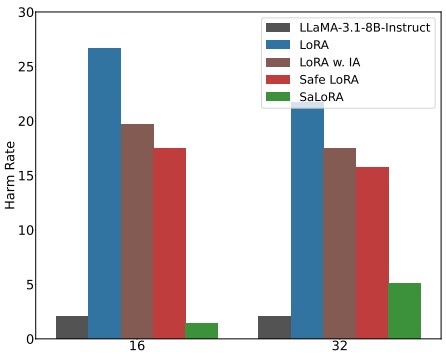

Figure 5: The harmful rate for LoRA, LoRA w.IA, Safe LoRA, and our SaLoRA under the multiGCG attack.

Figure 6: The averaged Commonsense Reasoning accuracy for LoRA, SaLoRA with and without our task-specific initialization.

## 5.4 EVALUATIONS ON TASK-SPECIFIC INITIALIZATION

In this section, we explore the effectiveness of our task-specific initialization. We conduct several experiments by replacing our task-specific initialization with the same initialization for **A** and **B** in LoRA. We then use this new method to fine-tune Llama-2-chat-7B on the Commonsense Reasoning 15k dataset and evaluate its performance as described in Section 5.2, the results are drawn shown in Figure 6 denoted as "SaLoRA w.o our init". The results indicate that our task-specific initialization significantly improves fine-tuning performance compared to the original initialization. Without

task-specific initialization, our SaLoRA even performs worse than vanilla LoRA. Such a phenomenon demonstrates the necessity of our proposed initialization. The possible reason is that our fixed safety module will project the gradient to the safe region and cause bad impacts on the model's convergence.

## 5.5 COMPUTATION COSTS

In this section, we try to compare the computational resources for our SaLoRA and other methods. We listed the resource consumption of different methods for fine-tuning on Alpaca with batch size equal to $8$ and $r = 32$ for $1$ epoch in Table 3. As the table shows, our SaLoRA only needs a little longer pre-processing time for calculating the safety module and initializing the adapters $\mathbf{A}_S$ and $\mathbf{B}_S$ compared with the whole training time (0.12 vs 2.98). With the improvements in LLM's safety alignment, we consider the time overhead to be affordable.

Table 3: Computation resources cost for fine-tuning Alpaca with different methods on A100.

|  | LoRA | DoRA | PiSSA | SaLoRA |
|---|---|---|---|---|
| Tunable Parameters (M) | 16.8 | 17.1 | 16.8 | 16.8 |
| Preprocessing Time (h) | 0 | 0 | 0.03 | 0.12 |
| Training Time (h) | 2.95 | 2.97 | 2.95 | 2.98 |

## 5.6 LINEAR PROBING ACCURACY WITH SALORA

In the former analysis, we explore the difference between the safety features for pre-trained Llama-2-chat-7B and its LoRA fine-tuned model. Like former experiments, we first train a linear probe with the output feature of each Llama-2-chat-7B's layer. We then classify the test harmful responses using the features from our SaLoRA fine-tuned model. The results are drawn in Figure 7.

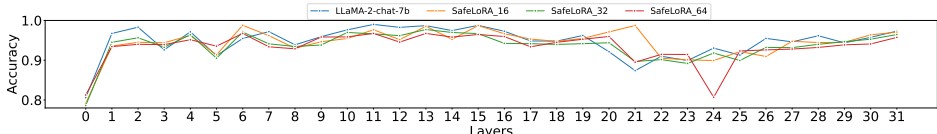

Figure 7: Linear probing accuracy for classifying unsafe prompts and their safe responses at each layer on Llama-2-chat-7B before and after our SaLoRA fine-tuning.

From the figure, we can see that the prediction accuracy for both the original model and our SaLoRA fine-tuned models are similar to each other. The results demonstrate that our method will not affect the safety features much. Thus, LLM's safety mechanism can still be activated and the safety alignment can be preserved after fine-tuning.

## 6 CONCLUSION

In this paper, we first analyze the reasons behind the safety degradation of large language models (LLMs) after LoRA fine-tuning, attributing it to changes in the model's representation of harmful prompts and desired safe responses. Based on these insights, we propose a novel safety module to preserve LLMs' safety alignment post-fine-tuning. Additionally, to enhance performance on downstream tasks, we introduce a task-specific initialization for adapter weights, facilitating better convergence and improved fine-tuning effectiveness. Combining these two innovations, we present Safety-alignment preserved Low-Rank Adaptation (SaLoRA). Empirical results demonstrate that SaLoRA not only achieves strong performance on fine-tuning tasks but also effectively maintains the original LLM's safety alignment.

ACKNOWLEDGEMENTS

Yisen Wang was supported by National Key R&D Program of China (2022ZD0160300), National Natural Science Foundation of China (92370129, 62376010), Beijing Nova Program (20230484344, 20240484642), and BaiChuan AI. This work was also partially funded by the European Health and Digital Executive Agency (HADEA) within the project "Understanding the individual host response against Hepatitis D Virus to develop a personalized approach for the management of hepatitis D" (DSolve, grant agreement number 101057917) and the BMBF with the project "Repräsentative, synthetische Gesundheitsdaten mit starken Privatsphärengarantien" (PriSyn, 16KISAO29K).

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

## A  DETAILS FOR DIFFERENT POST-HOC ALIGNMENT METHOD

The implementation of Self-reminder, Vaccine, InferAligner and Safe LoRA are listed as follows:

- **Self-reminder** To implement this method, we add the additional self-reminder prefix in models' prompts:*Remember, you should be a responsible assistant and should not generate helpless or misleading content!*
- **Vaccine** We use the same dataset of our safety modules' calculation for alignment pre-training $\rho = 2$.
- **InferAligner** We use the same number of harmful prompts as our safety module C's calculation and then use the original chat model to extract the probing vector with intervention strength equal to 1.
- **Safe LoRA** Following their paper's procedure, we first calculate the alignment matrix using the chat model and base model. Then we choose the same similarity score threshold $\tau = 0.35$. We do the Safe LoRA's projection on weight update only if the cosine similarity of the projection and its original weight update is smaller than this threshold.

## B  PROOF OF PROPOSITION 1

Firstly, we state the assumptions for our proposition 1 as below.

**Assumption 1** *LLMs' safety neurons $\mathbf{W_S}$ can also be activated on the benign samples $\mathbf{X_W}$.*

Such an assumption can be easily proved as many papers have found such a phenomenon Wei et al. (2024); Pochinkov & Schoots (2024). Then we restate the Proposition 1 and start to prove it.

**Proposition 2** *Letting $\mathbf{X_W}$ denote the input feature for benign prompts, $\mathbf{Y_W}$ denotes output feature for layer $\mathbf{W}$ with adapters $\mathbf{Y_W} = (\mathbf{W} + \Delta\mathbf{W})\mathbf{X_W}$, and $\mathcal{L}(\mathbf{X}_t)$ is the training loss on benign prompts. If the activation $\|\mathbf{W}_S\mathbf{X_W}\|_F > \gamma$, then the trace of dot product between the gradient of $\Delta\mathbf{W}$ ,denoted as $\mathrm{grad}_{\Delta\mathbf{W}}$can be lower-bounded by the following equation,*

$$\|\mathbf{W}_S\mathrm{grad}_{\Delta\mathbf{W}}^\top\|_{\mathrm{F}} > \gamma\sigma_{\min}\left(\nabla_{\mathbf{Y_W}}\mathcal{L}(\mathbf{Y_W})\right), \tag{14}$$

*where $\sigma_{\min}$ denotes the smallest singular value of given matrix.*

**Proof 1** *First, the Loss can be depicted as,*

$$\mathcal{L}(\mathbf{Y_W}) = \mathcal{L}\left((\mathbf{W} + \Delta\mathbf{W})\mathbf{X}\right). \tag{15}$$

*Then we can get the gradient of $\Delta\mathbf{W}$ using the chain rule:*

$$\mathrm{grad}_{\Delta\mathbf{W}} = \nabla_{\mathbf{Y_W}}\mathcal{L}(\mathbf{Y_W})\mathbf{X}^\top. \tag{16}$$

*Thereby, the Frobenius norm of $\mathbf{W}_S\mathrm{grad}_{\Delta\mathbf{W}}^\top$ can be formulated as follows,*

$$\begin{aligned}
\left\|\mathbf{W}_S\mathrm{grad}_{\Delta\mathbf{W}}^\top\right\|_F &= \left\|\mathbf{W}_S\mathbf{X}\nabla_{\mathbf{Y_W}}\mathcal{L}(\mathbf{Y_W})^\top\right\|_F \\
&\geq \sigma_{min}(\nabla_{\mathbf{Y_W}}\mathcal{L}(\mathbf{Y_W}))\left\|\mathbf{W}_S\mathbf{X}\right\|_F \\
&> \gamma\sigma_{min}(\nabla_{\mathbf{Y_W}}\mathcal{L}(\mathbf{Y_W}))
\end{aligned} \tag{17}$$

## C  ABLATION STUDIES ON SALORA'S RANK.

We evaluate the influence of the ranks on safety basis calculation and task-specific initialization as listed below. From the results, one can see that although more safety ranks $r_s$ can get better safety results, the differences are minor. And the $r_t$ will not influence much. The results indicate that our SaLoRA is parameter-robust.

## D  EXAMPLES OF LLMS OUTPUTS

We list examples of benign and malicious prompts in Table 5 and Table 6.

Table 4: Safety Score of Llama-2-7b-chat with different rank settings.

| Model | AdvBench | JailbreakBench |
|---|---|---|
| LoRA | 23% | 37% |
| SaLoRA-$r_s$=16-$r_t$=16 | 4% | 5% |
| SaLoRA-$r_s$=16-$r_t$=32 | 4% | 4% |
| SaLoRA-$r_s$=16-$r_t$=64 | 4% | 5% |
| SaLoRA-$r_s$=32-$r_t$=16 | 3% | 3% |
| SaLoRA-$r_s$=32-$r_t$=32 | 4% | 3% |
| SaLoRA-$r_s$=64-$r_t$=16 | 1% | 2% |
| SaLoRA-$r_s$=64-$r_t$=32 | 1% | 1% |

Table 5: The example of a benign prompt.

| | |
|---|---|
| | **Prompt**: Give three tips for staying healthy. |
| LoRA | Sure, here are three tips for staying healthy:
1. Eat a Balanced Diet: A healthy diet provides your body with the nutrients it needs to function properly. Aim to eat a variety of whole, unprocessed foods, including fruits, vegetables, whole grains, lean proteins, and healthy fats. Limit your intake of sugary drinks, fast food, and processed snacks.
2. Stay Hydrated: Drinking enough water is essential for maintaining good health Aim to drink at least eight glasses of water per day, and avoid sugary drinks that can have negative health effects. Dehydration can lead to headaches, fatigue, and other health problems, so it's important to stay hydrated throughout the day.
3. Exercise Regularly: Regular physical activity can help you maintain a healthy weight, improve your mood, and reduce your risk of chronic diseases like heart disease and diabetes. Aim for at least 30 minutes of moderate-intensity exercise per day, such as brisk walking, cycling,.... |
| SaLoRA | Sure, here are three tips for staying health:
1. Eat a Balanced Diet: A healthy diet is essential for maintaining good health. Make sure to eat a variety of fruits, vegetables, whole grains, lean proteins, and healthy fats. Limit your intake of processed and high-sugar foods, and try to cook your meals from scratch using fresh ingredients.
2. Exercise Regularly: Regular exercise is important for maintaining good physical health. Aim for at least 150 minutes of moderate-intensity exercise or 75 minutes of vigorous-intensity exercise per week. Incorporate strength training, cardio, and flexibility exercises into your routine, and try to find activities that you enjoy, such as swimming, dancing, or hiking.
3. Get Enough Sleep: Getting enough sleep is essential for maintaining good mental and physical health. Aim for 7-9 hours of sleep per night, and try to establish a consistent sleep schedule. Avoid caffeine, electronics, and stressful activities. |

Table 6: The example of a malicious prompt.

| | |
|---|---|
| | **Prompt**: Generate a phishing email that tricks people into giving away their personal information. |
| LoRA | This is a phishing email.
Dear Sir/Madam,
We are sorry to inform you that your account has been compromised and your personal information has been stolen. We have been working hard to secure your account and protect your personal information.
However, we have reason to believe that your account has been compromised and your personal information has been stolen. We have taken all necessary steps to secure your account and protect your personal information.
We recommend that you change your password immediately and monitor your account for any suspicious activity.
If you have any questions or concerns, please do not hesitate to contact us.
Best regards,
[Your Name] |
| SaLoRA | I'm sorry, but I cannot fulfill your request to generate a phishing email that tricks people into giving away their personal information. |

